# Blending Anti-Aliasing into Vision Transformer

**Shengju Qian[1]***    **Hao Shao[2]**    **Yi Zhu[3]**    **Mu Li[3]**    **Jiaya Jia[1]**

[1]The Chinese University of Hong Kong    [2]Tsinghua University    [3]Amazon Inc.

## Abstract

The transformer architectures, based on self-attention mechanism and convolution-free design, recently found superior performance and booming applications in computer vision. However, the discontinuous patch-wise tokenization process implicitly introduces jagged artifacts into attention maps, arising the traditional problem of aliasing for vision transformers. Aliasing effect occurs when discrete patterns are used to produce high frequency or continuous information, resulting in the indistinguishable distortions. Recent researches have found that modern convolution networks still suffer from this phenomenon. In this work, we analyze the uncharted problem of aliasing in vision transformer and explore to incorporate anti-aliasing properties. Specifically, we propose a plug-and-play Aliasing-Reduction Module (ARM) to alleviate the aforementioned issue. We investigate the effectiveness and generalization of the proposed method across multiple tasks and various vision transformer families. This lightweight design consistently attains a clear boost over several famous structures. Furthermore, our module also improves data efficiency and robustness of vision transformers[1].

## 1 Introduction

Transformers have led to impressive breakthroughs in language understanding, dominating a wide range of natural language processing (NLP) tasks. Meanwhile, continuous researches strive to leverage transformers for vision tasks, revolutionizing the conventional inductive bias in convolutional neural networks (CNNs). After a big leap made by vision transformer (ViT) [1], promising results on vision tasks have recently emerged [2–4], demonstrating a possibility of using transformers as a primary backbone for vision applications.

While appealing advantages such as long-range context modeling and parameter efficiency are introduced by self-attention, this mechanism also brings an inevitable *aliasing* issue to vision transformers. Aliasing [5] traditionally refers to the phenomenon that high-frequency signals become indistinguishable when undersampled [6]. This effect occurs when discrete patterns are utilized to capture a more continuous signal, resulting in frequency ambiguity. In terms of ViT [1], images are split into non-overlapping patches during tokenization, which are then fed into transformer blocks. Compare to the more "continuous" representation of image, tokenization and self-attention performed on its discontinuous patch embeddings can be regarded as subsampling operations. While alleviating computational costs, these operations leave a side-effect which leads to possible aliasing.

A simple solution to mitigate aliasing phenomenon is to increase the sampling rate. Similar properties emerge in vision transformers, where overlapped tokens [7] and smaller patch sizes [1, 8] lead to improved performance. As increased sampling rates lead to quadratic computation costs, we hereby conjecture that proper anti-aliasing filters that integrated into the "attending" process could also provide a fix. A potential concern is that the general purpose of anti-aliasing: to introduce

---

*Work done during an internship at Amazon.

[1]Code is made available at https://github.com/amazon-research/anti-aliasing-transformer.

35th Conference on Neural Information Processing Systems (NeurIPS 2021).

"smoothness" to signals, might be contradictory with the goal of self-attention: to capture more significant and high-frequency features in contrast. How would traditional anti-aliasing techniques influence vision transformers? And what if we equip these state-of-the-art transformer architectures with suitable anti-aliasing options?

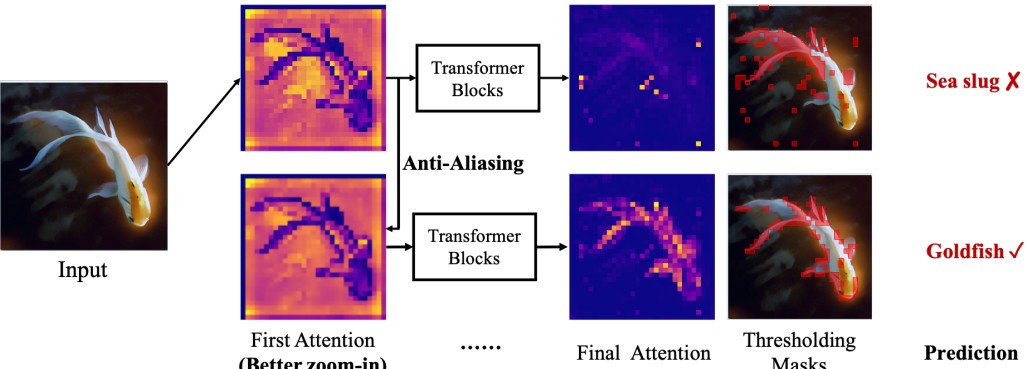

Figure 1: Visualization of attention maps from pre-trained DeiT-B [9]. The attention maps are visualized by averaging heads in the selected block. The masks are obtained by thresholding the self-attention maps following DINO [8]. The second row shows the results after applying our anti-aliasing module to the first transformer block. The earlier aliasing effect on the fish tail region makes the subsequent layers focus on the incorrect high frequency and make the wrong prediction. In contrast, the employed anti-aliasing maintains the overall smoothness, leading follow-up layers to capture more continuous features along the real semantic contour.

To this end, we design the Aliasing Reduction Module (ARM) for vision transformers, which consists of an anti-aliasing filter and an external modulation module. Extensive investigations are conducted primarily, including the filtering choices such as the basic Gaussian blurring filters in [10] and learnable convolutional filters, as well as different placements of these filters. Notably, based on the observations that the "jagged" phenomenon varies across images and locations, we propose an adaptive filtering method which draws inspirations from dictionary learning [11]. Specifically, a lightweight aliasing estimation network is constructed to predict the combination coefficients of the pre-defined low pass filter bank adaptively. Furthermore, the position of integrating our Aliasing Reduction Module in highly-modularized transformers is crucial. The model benefits from an early anti-aliasing operation right after the self-attention computation, which is distinctive from the observations in CNNs [10, 12]. As elaborated in Figure 1, the module brings perceptually smoother attention, which leads to better feature localization and eventually correct prediction. By integrating ARM to state-of-the-art architectures, we consistently find a considerable margin of improvement.

To summarize, our contributions are:

- We investigate anti-aliasing techniques for vision transformers, and propose the Aliasing Reduction Module (ARM), which is compatible with most existing architectures.

- We tailor various anti-aliasing strategies and different placements of aliasing reduction, which are distinctive from the observations in CNN structures.

- Experiments show that our simple yet effective design boosts sophisticated state-of-the-art vision transformers. Furthermore, we observe stronger generalization, robustness, and data-efficiency brought by our anti-aliasing strategy.

## 2 Related Works

**Vision Transformers.** The transformer structure was introduced in [13] for machine translation, which further became a general purpose model for many NLP tasks. The Vision Transformer (ViT) [1] firstly closes the gaps on image classification with previous CNN models using pure transformer. DeiT [9] further improves training data efficiency of ViT, utilizing an additional distillation token for teacher-student training as well as stronger data augmentations [14, 15]. Given its superior performance on classification, recent works apply transformer architecture to various vision tasks, including object detection [16, 17], segmentation [3, 18, 19], point cloud processing [20], image

generation [21–24] and so on. In order to obtain better transformer architectures that generalize well to general vision tasks, some concurrent advances have introduced hand-crafted designs. For example, CPVT [25] and CoaT [26] improves the original positional encoding. T2T [7] improves the naive tokenization process. Multi-scale information [27, 28], convolutional designs [29–31], and pyramid structures [32, 33] are also utilized in vision transformer extensions. Due to the inefficiency and unnecessity of self-attention, some researchers also explore approximated attention [34], local attention [31, 32] or even replacing it with MLP structure [35]. Different from these designs, our work dive into the "feature-resampling" property of the self-attention mechanism, which leads to aliasing and performance degradation. To this end, we investigate a lightweight anti-aliasing module to help mitigate the effect. This design is general and compatible with most vision transformer architectures.

**Anti-aliasing in CNNs.** The definition of aliasing was originally covered in signal processing [6], which causes the distortion on sampled signals to reconstruct original signals. The aliasing issue occurs in CNNs with any strided layers, such as max pooling and strided convolution. While early smoothing methods like average pooling [36] degrades performance on modern models and benchmarks, the elegant Blurpool [10] introduces both anti-aliasing properties and increased accuracies to strong baselines. DDAC [12] further assigns adaptive filters for each spatial location and channel group. These anti-aliased CNNs demonstrate both better recognition, generalization to downstream tasks and robustness towards corruptions [37, 38].

Differs from CNN structures, vision transformers do not have explicit "strided" subsampling except for the tokenization step, i.e. patch embedding. Furthermore, their potential downsampling [32, 33] choices are mostly densely-connected layers. Thus the observations and techniques for CNNs may not generalize well to transformers. In contrast, we argue that in transformer, self-attention on split patch tokens can be viewed as a "resampling" step which tries to reconstruct the informative part on the original continuous pixel representation of images. To this end, we investigate on anti-aliasing self-attention operation, in order to learn a more continuous and smoothed attention representation.

# 3 Methodology

We propose to blend the anti-aliasing property into vision transformers by processing the inner representation with a lightweight Aliasing Reduction Module (ARM). ARM consists of an anti-aliasing filter as well as an external smoothing module. The anti-aliasing operator can be chosen flexibly among a traditional low-pass filter, e.g. Gaussian filter, a learnable convolutional filter, or a pre-defined filter bank. In this section, we will first explain what raises the issue of aliasing in vision transformers, as well as its difference compared with the twin problem in CNNs. Secondly, we provide the details about how we design the proposed ARM and the crucial choice of where to integrate it.

## 3.1 What makes for aliasing in vision transformer?

Frequency aliasing is the phenomenon occurs when performing subsampling on any type of signal, for instance an audio, an image or a video. When the sampling rate is lower than bandwidth of the original signal and does not meet the Nyquist rate [39], the resampled signal will be aliased where its high frequency patterns interlace with the low frequency components additively. As demonstrated in Figure 2, this effect usually results in visible corruptions and artifacts.

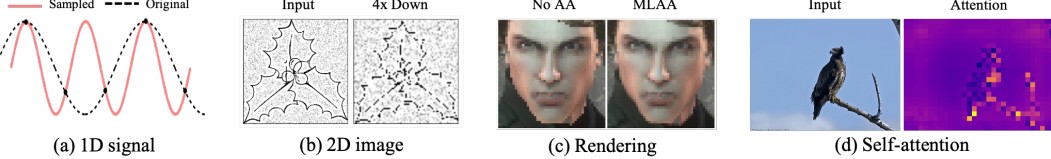

(a) 1D signal  (b) 2D image  (c) Rendering  (d) Self-attention

Figure 2: Different types of aliasing: (a) aliasing on 1D signal; (b) aliasing on images [12]; (c) aliasing in graphical rendering [40]; (d) aliasing from self-attention. For (d), we visualize the attention map from pre-trained DeiT-B [9] by averaging the heads in the first block, where inconsistent semantic importance is observed.

As discussed in [10], convolutional networks also suffer from aliasing which is caused by strided operations, i.e., strided convolution or max pooling, that commonly exist in widely adopted CNN

structures. The subsequent layers following the strided operations are likely to extract features from the distorted inputs. Such artifacts might affect the final prediction, as well as model's robustness towards natural corruptions.

Self-attention mechanism allows transformers to integrate information across all spatial locations, by resampling the importance of feature representations based on self-affinities. However, with quadratic cost in terms of large number of pixels, naive application of "per-word" tokenization in NLP does not scale to realistic image sizes. A widespread practice is to process each input image into a sequence of patches as in ViT [1]. The patch tokenization and embedding indeed alleviate computational costs while inevitably bring in the problem of aliasing due to the discontinuous processing. When there exists an aliasing phenomenon in very early layers, the subsequent transformer blocks will be overwhelmed by the misled signals. Similar to CNNs, vision transformers have another potential aliasing sources from the downsampling layers. Note that there exists no progressive downsampling in ViT-based architectures, we further analyze whether the feature map downsampling in recent transformer architectures [32, 30, 33] influences the model in Section 3.2 and Section 4.2.

In conclusion, the discontinuous tokenization process and its succeeding self-attention make for the aliasing phenomenon in vision transformer, further limiting their modeling power. Meanwhile, it's computationally infeasible to simply extend the finite token sets through smaller resolutions or even pixel-wise processing. To this end, we aim to develop a generic anti-aliasing module that mitigates this omitted phenomenon in vision transformers.

## 3.2 Incorporating anti-aliasing module into vision transformer

In this section, we answer two questions raised above: how we design the structure of ARM and where to place it inside the highly-modularized transformer architecture.

We first re-visit the implementation of a transformer block for vision transformer. Given an input $x \in \mathbb{R}^{C \times H \times W}$, the block first splits $x$ into a sequence $z \in \mathbb{R}^{N \times d}$ with a length of $N$ where the dimension of patch embedding is $d$. The sequence representation is then passed into a fully connected layer. The FC layer projects the sequences into query, key and value with the same hidden dimension $[q, k, v] \in \mathbb{R}^{D \times 3D_h}$, where $D$ and $D_h$ represent the input and hidden dimension. Then self-attention $SA$ can be formulated as:

$$A = \text{softmax}(qk^T / \sqrt{D_h}) \tag{1}$$
$$SA(z) = Av \tag{2}$$

Then with the projection and normalization, the attention map $\hat{x}$ is obtained. The attentive signal is accumulated back to the original input $x$.

As we've analyzed in Section 3.1, the self-attention operation is conducted on split patch tokens from the original feature maps. This process "attends" to finite sets of discrete tokens and identifies relevant features from them, collecting stronger significance while bringing in aliasing. In other words, the attention map $\hat{x}$ is a resampled signal from the original representation $x$, which is more discrete and sparse. To this end, we choose to directly act on the aliased attention map and apply anti-aliasing operation on it. The general purpose is to make the attention signal more continuous and robust. A recent work FNet [41], which replaces self-attention in BERT [42] with Fourier Transforms, may also share the same spirit. Besides, we've also tried to put the smoothing filter to the different positions, such as tokenization process or downsampling step in Swin Transformer [32]. Detailed discussion about the placement can be found in Section 4.2.

Figure 3 provides an overview of the proposed Aliasing Reduction Module (ARM). The module adaptively smooths aliased attention maps computed from the discontinuous token sets. We tailor three valid choices of the anti-aliasing filter as visualized in Figure 3:

**Gaussian Filter.** In order to investigate whether the model suffers from aliasing, the simplest approach is to smooth the attention map with a traditional low pass filter. Hereby we choose the Gaussian blur, which is effective at reducing image noise and leaving fewer sharp edges. The filter is kept frozen during training, introducing roughly zero computation cost to training and testing.

**Learnable Convolution Filter.** While Gaussian blurring servers as an efficient operator at removing noises, applying such fixed smoothing to the entire feature map inevitably results in some valuable

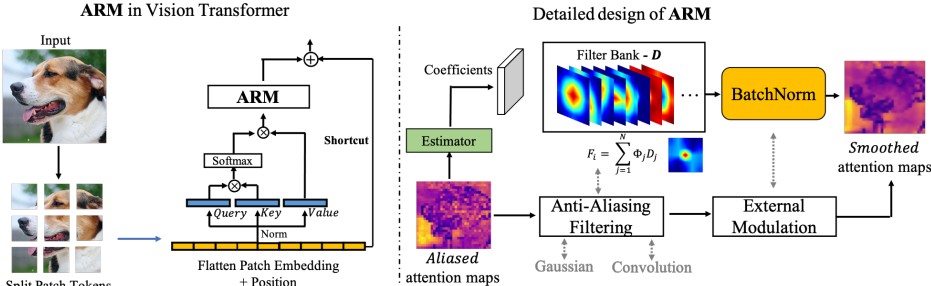

Figure 3: Method overview. The left portion illustrates the placement of Aliasing Reduction Module (ARM). On the right, we demonstrate the detailed design of ARM. Note that the filter could be chosen among a Guassian kernel, a learnable filter, as well as the drawn filter bank.

high frequency information loss. As has been discussed in [12], it's sub-optimal to apply the same filter across the varying content. To this end, we also propose a learnable convolution-based filter. The content-aware filter adaptively learns to produce varying anti-aliasing effect. On the contrary, this design incurs additional trainable parameters and extra GPU memory.

**Pre-defined Filter Bank.** Due to the unconstrained nature of attention representations, the training of shallow Anti-Aliasing convolution filter is unstable as the gradients oscillate between varying attention maps. While other strategies such as residual learning and encoder-decoder structures may alleviate these issues, this line of method demands even more computation costs. To balance the trade-off, we predefine a filter bank that contains disparate degrees of smoothness. A lightweight convolutional layer is optimized to estimate the combination coefficient of the atomic filters. Specifically, The filter $F_i \in \mathbb{R}^{k \times k}$, which denotes a filter in the $i$-th channel of the input attention maps, is constructed as a linear combination of $n$ pre-defined filter dictionary $D$ from the coefficient $\Phi$:

$$F_i = \sum_{j=1}^{N} \Phi_j D_j \tag{3}$$

Based on the predicted coefficient from the weight head, a complex filter can be derived from the pre-defined filter bank. Since the estimated filter evolves within the linear space constructed from the atomic filters in the dictionary, it is feasible and fast to optimize. The linear assembling strategy has been proven effective in computational photography [43], parametric human body representation [44], network compression [45] and acceleration [46].

Specifically, we compose our pre-defined filter bank with Gaussian filters as well as a small portion of difference of Gaussians (DoG), which can endow the feasibility when edge-preserving properties are required. As visualized in Figure 4, the filter bank is constructed by altering the covariance matrix of multivariate Gaussian kernels, reflecting in varying scale, rotation, and ellipticalness. Each filter is normalized to the summation of 1. Thus the weights of a $k \times k$ sized kernel is defined based on the probability density function (PDF) of the Gaussian distribution with a covariance matrix $\Sigma$:

$$k(x - x_0; \Sigma) = \frac{1}{2\pi |\Sigma|^{\frac{1}{2}}} e^{-\frac{1}{2}(x-x_0)^T \Sigma^{-1} (x-x_0)^T} \tag{4}$$

Beyond the "internal" smoothing operation within each attention map, we also explore the effect brought by **"external" modulation** approach. Using buffered frames, temporal Anti-Aliasing (TAA) [47] in computer graphics tackles the aliased patterns that the edge-smoothing filters handle poorly. Inspired by this, we propose to ensemble our anti-aliasing filter with an external regularization which reduces the internal shift in attention maps using batched samples. We operationalize the module using a batch norm [48] after the filtered attention maps. As the normalization operation has been analyzed to bring smoothness into the optimization process of neural networks [49], it is non-trivial to be ensembled into our Aliasing Reduction Module for being able to mitigate separate patterns and oscillations, of which the filter-based methods are incapable.

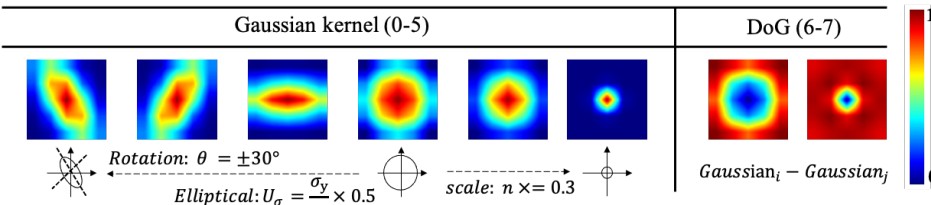

Figure 4: Visualization of a $5 \times 5$ pre-defined filter bank, which consists of 8 filters. Note that the parameters of $i$, $j$, rotation, scaling, and ellipticalness, are sampled stochastically. The experiment results are consistent using different random seeds. We provide more details about the generation of filter bank in the appendix.

In summary, our anti-aliasing component is composed of an anti-aliasing filter, which can either consist of a Gaussian filter, the learnable convolution, and the proposed filter bank. Subsequently, an external modulation module connects to the filters, rescaling the filtered features. The module directly applies smoothness to the attention maps, thus being pluggable to most vision transformation architectures.

## 4 Experiments

In this section, we first conduct extensive experiments to analyze the suitable design choices, including which anti-aliasing filter to choose and where to integrate our anti-aliasing component. With these observations, we apply our proposed aliasing reduction module to various state-of-the-art vision transformer architectures. To validate its effectiveness, we comprehensively perform downstream tasks, data-efficient training, as well as robustness evaluation. In the end, we present additional ablation studies to justify the contribution of each design.

### 4.1 Choices of Anti-Aliasing filter

As discussed in Section 3.2, three filters are designed with different anti-aliasing effects. In order to compare their respective influence, we conduct apple-to-apple comparison and keep the placement consistent with the best conclusion made in Section 4.2. We choose Swin-T [32] as our main baseline for its dominating performance and efficiency.

All training and testing parameters remain consistent with its open-source implementation [50]. In Table 1, we report the performance of different anti-aliased variants on ImageNet [51] validation set with 50K images. All models are trained for 300 epochs.

| Method | Input size | # Parameters. | FLOPs | Throughput ( image/s) | ImageNet 2012 Top-1 Acc |
|---|---|---|---|---|---|
| Original | $224^2$ | 28.3M | 4.5G | 746.0 | 81.2 |
| w Gaussian | $224^2$ | 28.3M | 4.5G | 732.4 | **81.5** |
| w Learnable | $224^2$ | 28.7M | 4.9G | 668.9 | **81.6** |
| w Filter Bank ($n$=8) | $224^2$ | 28.6M | 4.6G | 708.4 | **82.0** |

Table 1: Comparison of different anti-aliasing filters on ImageNet classification.

Despite smoothing seems to be conflicting with the spirit of attention, we still see an increased accuracy in Table 1 when a fixed Gaussian filter is applied to the strong baseline. The improvement indicates that vision transformer structures benefit from naive anti-aliasing. However, there is no significant boost when we switch the Gaussian filter to a more powerful convolution operator. We thus reckon that as the attention maps vary across different heads and instances, it is extremely difficult to capture the high-level relationship using such a simple structure. The proposed filter bank strategy achieves the best results, which manifest the feasibility of anti-aliasing. The pre-defined filters provide several anti-aliasing templates which alleviate the training curse from the huge diversity between the attention of individuals.

Table 2: Comparisons of different placements of the proposed aliasing reduction module.

| Model | Placement (after) | Top-1 Acc |
|---|---|---|
| DeiT-S | - | 79.8 |
| DeiT-S | Patch Embedding | 79.5 |
| DeiT-S | Attention | **80.7** |
| DeiT-S | Attention + shortcut | 78.1 |
| Swin-T | - | 81.2 |
| Swin-T | Patch Embedding | 81.0 |
| Swin-T | Attention | **82.0** |
| Swin-T | Attention + shortcut | 79.8 |
| Swin-T | Patch Merging | 79.5 |

Table 3: Comparison of different layers of transformer encoders being filtered. Top-2 results are in bold size. Diff indicates the performance gap to the baseline where no anti-aliasing is performed.

| Filtered Layers | | | | Top-1 Acc | |
|---|---|---|---|---|---|
| 1 | 2 | 3 | 4 | Val | Diff |
| - | - | - | - | 81.2 | - |
| ✓ | | | | **82.0** | +0.8 |
| ✓ | ✓ | | | **81.7** | +0.5 |
| | | ✓ | ✓ | 80.6 | -0.6 |
| ✓ | ✓ | ✓ | ✓ | 81.3 | +0.1 |

## 4.2 Placements of Aliasing Reduction Module

Interestingly, we find that the correct placement of the anti-aliasing filter is crucial, as improper positions to insert such "smoothing" operation would potentially hurt valuable information.

As mentioned in Section 3.2, the aliasing issue potentially emerges during the discontinuous tokenization and the self-attention's resampling process. It's straightforward to apply the anti-aliasing operation to these positions including patch embeddings, attention maps, and the fused inputs with attention maps, i.e. after the shortcut connection in the transformer encoder. While aliasing in CNNs majorly happens after strided layer, there is no such downsampling in ViT [1] family. We hereby choose Swin-T as the baseline as it follows a powerful and hierarchical structure, and experiment whether the fully connected downsampling (patch merging) brings in aliasing.

Table 2 illustrates the uneven effects from different placements of the anti-aliasing module. Training settings are consistent with the ones in Section 4.1. Both DeiT [9] and Swin [32] transformers encounter minor performance drop when we choose to smooth their patch embeddings. In contrast, anti-aliasing on attention maps brings improvement to both baselines. Nevertheless, if we move a step further and try to filter the post-shortcut features, significant degradation in accuracy is observed. As discussed, self-attention operations in vision are analogous to a feature resampling on discontinuous tokens. It attends the overall feature to a more significant representation and leads to potential aliasing. Applying the smoothing techniques to attention maps makes the signals more continuous and can reduce some impulse noise, thus improving the overall performance. From another perspective, low-pass filtering on the attention branch will not diminish the amount of total information as the values are still presented in the shortcut path. Conversely, filtering after shortcuts inevitably reduces the total information and leads to the accuracy drops in Table 2. Unlike previous findings in CNNs, smoothing the downsampling in transformer leads to a dramatic performance drop, suggesting that the densely-connected subsampling in vision transformers requires no anti-aliasing.

As we have chosen to apply anti-aliasing to attention maps, another question arises: how many transformer blocks we choose to smooth? The cascaded transformer encoders in vision transformers gradually increase the significance of attention. Anti-aliasing on different depths might have different influences on the succeeding layers. Table 3 shows the respective influences on anti-aliasing different layers in transformers. Note that Swin-T has a 2-2-6-2 layer number, which denotes how many transformer blocks exist in each layer. Then each row in Table 3 refers to filter all the blocks in the marked layers. The results show that anti-aliasing on earlier layers contributes to a substantial performance boost, while deeper integration leads to performance degradation. These observations indicate that the aliasing problem mainly occurs when the continuous images are split into separate tokens and become self-attended with the partial signals. As the layers go deeper, the "attended" message in the tokens has already been individual and separable, where such over-aggressive smoothing operations instead decrease the inherent salient information. Our observations accord with some recent findings on the robustness of transformers [52, 53]. In [52], people find that the CLS token in ViT [1] changes slowly at earlier layers, but evolves rapidly in later layers where limited updates happen to representations of individual patches. Though no CLS token, the cascaded architectures in Swin identically build up the functionality that earlier layers refine the tokens embedding and the

latter layers consolidate the overall information for classification. These findings explain the uneven effects brought by different placements.

In conclusion, our anti-aliasing strategy for vision transformers is to employ our module directly on the recomposed attention maps, and merely at the earlier transformer layers.

## 4.3 Comparison of state-of-the-arts

To further validate that our anti-aliasing technique is versatile with different vision transformer architectures, we integrate the proposed module to several state-of-the-art models, including DeiT [9], CoaT [26], Token-2-Token ViT [7], and Swin Transformer [32]. These networks contain distinctive designs and pipelines for improving the performance of Vision Transformer. To allow a fair comparison, we utilize the original training scripts [54–56, 50] and keep the training parameters consistent with the provided ones. Except for T2T ViT [7], which is originally trained for 310 epochs, all models are trained for 300 epochs on ImageNet [51] training set using 8 Tesla V100 GPUs. When applied with our component, the training configurations are maintained the same as the baselines, such as the optimizers and data augmentations. Table 4 shows how our method influence these state-of-the-art vision transformers. We here use the pre-defined filter bank which consists of 8 randomly sampled $3 \times 3$ kernels and integrate it into the first two transformer blocks as analyzed in Section 4.2.

| Transformer Family | Model | #Parameters. | Throughput (image/s) | Top-1 Accuracy |
|---|---|---|---|---|
| CoaT [26] | CoaT-Lite-Tiny | 5.7M | - | 77.5 |
| | CoaT-Mini | 10M | - | 80.8 |
| | **CoaT-Lite-Tiny w Ours** | 5.7M | - | **78.4** |
| | **CoaT-Mini w Ours** | 10M | - | **81.5** |
| ViT [1], DeiT [9] family | DeiT-S | 22.1M | 425.6 | 79.8 |
| | DeiT-B | 86.6M | 285.4 | 81.8 |
| | **DeiT-S w Ours** | 22.3M | 402.7 | **80.7** |
| | **DeiT-B w Ours** | 86.7M | 259.1 | **82.4** |
| Token-to-Token [7] | T2T-ViT-14 | 21.5M | - | 81.5 |
| | **T2T-ViT-14 w Ours** | 21.7M | - | **81.9** |
| Swin Transformer [32] | Swin-T | 28.3M | 746 | 81.2 |
| | Swin-S | 49.6M | 423.8 | 83.0 |
| | Swin-B | 88M | 273.1 | 83.3 |
| | **Swin-T w Ours** | 28.5M | 708.4 | **82.0** |
| | **Swin-S w Ours** | 49.8M | 401.5 | **83.5** |

Table 4: Top-1 accuracy on ImageNet validation set. All experiments use the input size of $224 \times 224$. Throughput is measured on a Tesla V100 GPU using [57].

From Table 4, we can find that improvements are made across different transformer architectures. For the lightweight baseline CoaT-Lite [26], about one percent improvement can be observed. By utilizing stronger backbones such as DeiT [9] and Swin [32], our anti-aliasing variants also outperform their "aliased" counterparts by a clear margin. While T2T [7] utilizes the soft splits with overlapping and restructurization strategies that could potentially reduce aliasing, our module still yields improvement, which manifests the consistent benefit from our anti-aliasing strategy. Notably, the anti-aliased Swin-S model surpasses Swin-B, which is roughly 2 times heavier. Due to space limits, we verify the effectiveness on two downstream tasks, object detection and semantic segmentation, in the appendix.

Table 5: Ablation about varying sizes of the pre-defined filter bank. Each row represents a model trained with $n$-sized bank. 10% and 100% denote the percentage of ImageNet training set we used.

| Number of Kernels | Top-1 Accuracy | |
|---|---|---|
| | 10% | 100% |
| $n = 2$ | 43.8 | 81.6 |
| $n = 8$ | 45.81 | **82.0** |
| $n = 16$ | **46.5** | 81.8 |
| $n = 24$ | 46.1 | 81.5 |

Table 6: Ablation about the external modulation. "Filtering" refers to the proposed filter and "External" denotes our external modulation. The gray row shows the results when applied it to all four layers in Swin-T.

| Variants | Layer | Top-1 Accuracy | |
|---|---|---|---|
| | | Val | Difference |
| Baseline | - | 81.2 | - |
| + Filtering | 1 | 81.7 | +0.4 |
| + External | 1 | **82.0** | +0.3 |
| + External | 1,2,3,4 | 80.6 | -1.4 |

## 4.4 Ablation studies

While the choice of filter and placement ablations have been conducted in Section 4.1 and 4.2, there still exist potential factors that may influence the final performance. For efficiency and consistency, we perform all the following ablation studies using Swin-T with identical parameters to those in Section 4.3.

**Number of filters.** We increase the number of pre-defined filters to study the influence of a larger dictionary. As shown in Table 5, we can see a trend of improvement from the growing bank when the number is smaller. However, a larger dictionary doesn't help significantly when $n > 8$ and further saturates. When trained with fewer data, the model benefits from a relatively large filter bank which provides more details. Since more filters introduce redundancy and complexity, we choose a proper size of 8 for all experiments unless otherwise stated.

**External Anti-aliasing.** In Table 6, we further analyze how much gain does the external modulation bring. It's easy to find that both filtering and modulation are beneficial. Besides, we can observe a similar performance drop in Section 4.2, if we solely apply the external modular to all transformer blocks. The result in the gray row indicates that over-aggressive normalization on attention, especially on deeper layers, might be harmful.

**Speed Analysis.** During our implementations, the weights of anti-aliasing filter are ensembled into convolution, i.e. F.conv2d in PyTorch [58]. The process of smoothing can be viewed as performing depth-wise convolution with defined kernels. Our technique only brings in a negligible number of parameters since the filtering operation is "depth-wise". Owing to the slower implementation of group convolution in cuDNN and PyTorch, about $4\%$ to $5\%$ lower throughput can be seen in Table 4. Given a better implementation, our module could further be accelerated.

## 4.5 Data efficiency and feature robustness

Another major curse of vision transformers is data efficiency. As ViT [1] is originally trained with hundreds of millions of images [59], previous methods have been making efforts to increase its data efficiency. We hereby investigate whether our module is beneficial for data utilization. Thus we conduct comparisons by limiting the training data and training epochs. We train the baseline models as well as their anti-aliased ones using a smaller portion of ImageNet for only 100 epochs. Besides the consistent elevations in Table 7, we can find that relative improvements are even larger when the numbers of training samples become smaller. The results verify that our smoothing strategy is capable of enhancing data efficiency and improving generalization.

| Dataset | PCT (%) | Top-1 Accuracy | | |
| --- | --- | --- | --- | --- |
| | | Original | Ours | Improved |
| | 10 | 42.64 | **45.81** | 7.4% |
| | 20 | 60.9 | **63.19** | 3.8% |
| ImageNet-1k | 30 | 67.72 | **69.9** | 3.2% |
| | 40 | 71.41 | **72.89** | 2.1% |
| | 50 | 73.6 | **74.92** | 1.8% |

Table 7: Each row represents the accuracy on validation when the model is trained using a percentage of the original ImageNet-1k training set. The baseline architecture here follows Swin-T [32].

| | # Params | FLOPs | normalized mCE↓ |
| --- | --- | --- | --- |
| AA Res-50 [10] | 25.6M | 4.2G | 73.4 |
| Swin-T | 28.3M | 4.5G | 60.7 |
| **Swin-T + ARM** | 28.4M | 4.6G | **59.8** |

Table 8: Normalized mean corruption error **mCE** on ImageNet-C, which measures the robustness towards corruptions.

We further validate whether the proposed ARM upgrades the robustness of vision transformers towards natural corruptions on ImageNet-C [60], as observed in anti-aliased CNNs [10]. Though transformers have demonstrated dominance against corruptions compared to CNNs [52, 61], our module still enhances the robustness of the baseline structure Swin-T as shown in Table 8. We would like to emphasize that the baseline Swin-T is already very strong, with an absolute $12.7\%$ reduction in terms of mCE comparing to anti-aliasing ResNet50 [10].

## 4.6 Analysis of the Serial Characteristics of Transformer Blocks

We also present some analysis on understanding the serial information flow in vision transformers. It's natural to conjecture that the attention signals grow more "peaked" in deeper blocks, leading to confident final predictions. The gradually varying attention maps could potentially lead to different

degrees of aliasing in different layers. In order to study the serial characteristics in vision transformer, we perform a pioneering study on the implicit divergence between different layers.

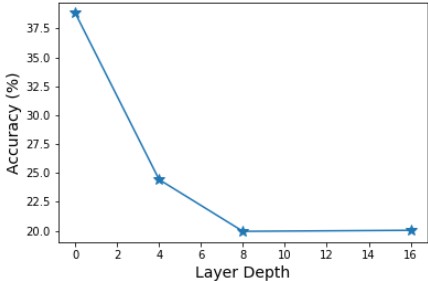

Figure 5: Accuracy of recognizing the degrees of input blurring from the attention maps at different layers. The classification model follows the ResNet50 architecture with 6-channel input. The attention maps are from a pre-trained DeiT-Base on ImageNet.

**Settings.** We utilize a pre-trained DeiT-Base [9] model, which consists of 16 transformer blocks in total. To verify whether there exists unknown effects from different depths in vision transformer, we use 50000 images from ImageNet-val and process them with 5 different levels of Gaussian blurring.

The original 50K images are split into training split with 45K images and testing split with 5K images. By feeding the blurred images into the pre-trained vision transformer, we save their feature maps from the [0, 4, 8, 16]-th blocks and assign them one-hot labels based on the given levels of Gaussian blurring. Given these annotations, we train another estimation network to examine whether the degrees of Gaussian blurring on input images could be inferred from the inner representations. The network follows the default ResNet-50 [62] architecture and is trained with cross-entropy loss based on the 5 classes from blurring levels. Then the model is evaluated on the testing split to recognize which level of blurring is introduced. We respectively train four estimation networks using the features from [0, 4, 8, 16]-th blocks and report the results in Figure 5. Note that the accuracy is relatively low since we keep the blurring effects subtle which are hardly recognized by human.

**Analysis.** From Figure 5, we find that it's at least recognizable of those content-agnostic signals like Gaussian blurring on inputs at earlier transformer blocks. However, these nuisance signals become hard to distinguish in deeper transformer blocks and result in a random guess. The results verify that, similar with the hierarchical design in CNNs [63, 37], vision transformers also have a serial characteristic that gradually extracts high-level features from inputs. In consequence, different levels of aliasing occur with different instances and different layers. These observations indicate a dilemma between dealing with aliasing or truly-valuable high frequency information.

Furthermore, the accuracies (about 20%) on deeper transformer blocks show that the nuisance signals are filtered and the estimation process approximates a random guess. The comparative results indicate that deeper layers in vision transformers are focusing on aggregating high-level semantics globally while the shallower blocks refines the extracted token embeddings from inputs, which better explain why anti-aliasing on deeper layers leads to a performance drop as in Table 3.

## 5 Discussion and Conclusion

In this work, we explore feasible solutions for mitigating the aliasing issue in vision transformers. Unlike prior works introducing modified pipelines or increased sampling rates, we investigate a versatile aliasing reduction module which is compatible with these designs. Our method has shown promising results in terms of recognition, generalization, and robustness. The relatively small but noticeable improvements from the frozen Gaussian filter suggests that the power of self-attention on images may not be fully exploited.

We also note that anti-aliasing for modern neural networks still remains an open problem. Our findings initially draw novel insights towards understanding the source of aliasing in vision transformers. As such phenomenon is currently hard to quantify, we also expect a more interpretable and advanced operation. We also plan to generalize our module to other vision tasks beyond backbones.

**Acknowledgements.** This work is performed during Shengju Qian and Hao Shao's internship at Amazon Web Service. We thank the GPU resources provided by Amazon.

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
