# 1 Appendix

2 The content of appendix is organized as follows:

- 3 • Appendix A provides more details about the generation process of filter bank.

- 4 • Appendix B shows the Pytorch-style code of our proposed ARM.

- 5 • Appendix C conducts evaluations on downstream tasks including object detection and se-
- 6 mantic segmentation. We also include more details about the robustness towards corruptions.

## A  Details about the generation process of filter bank

8 In this section, we provide more details about how we generate the filter bank. The filter bank
9 consists of Gaussian and Difference of Gaussians (DoG) filters. As mentioned, the Gaussian filters
10 are designed for its widely-adopted ability in anti-aliasing [1, 2] and image enhancement [3], while
11 Difference of Gaussians can boost the power of edge-aware operations [4].

12 In generating filters, the weights of each $k \times k$ kernel inside the filter bank are defined according to
13 the function below:

$$k(x - x_0; \mathbf{\Sigma}) = \frac{1}{2\pi |\mathbf{\Sigma}|^{\frac{1}{2}}} e^{-\frac{1}{2}(x-x_0)^T \mathbf{\Sigma}^{-1}(x-x_0)^T} \tag{1}$$

14 In particular, the weights are generated based on the covariance matrix $\mathbf{\Sigma}$ which can be decomposed
15 into the equation:

$$\begin{aligned} \mathbf{\Sigma} &= \gamma^2 \mathbf{U}_\theta \mathbf{\Lambda} \mathbf{U}_\theta{}^T \,, \\ &= \gamma^2 \begin{bmatrix} \cos\theta & -\sin\theta \\ \sin\theta & \cos\theta \end{bmatrix} \begin{bmatrix} \sigma_1^2 & 0 \\ 0 & \sigma_2^2 \end{bmatrix} \begin{bmatrix} \cos\theta & -\sin\theta \\ \sin\theta & \cos\theta \end{bmatrix}^T \,, \end{aligned} \tag{2}$$

16 where the rotation, scaling, and elongation (ellipticalness) parameters are represented by $\theta, \gamma$, and
17 $\sigma_{1,2}$, respectively. During each run, we sample these parameters in intervals stochastically. After
18 generating the covariance matrix $\Sigma$ for each Gaussian filters, we also sample groups of $i, j$ to generate
19 the weights of DoG filters by subtracting the derived Gaussian kernels $i$ and $j$.

20 During our implementations, we find that the results are robust towards different parameters. One
21 major reason is that the filter bank is redundant and contains enough representation power. When
22 sampled with different random seeds, the estimator is capable of generating abundant kernels.
23 However, we also observe the oscillations of accuracy (about 0.5% Top-1 accuracy variance in 10
24 runs) when the small portion of DoG filters are removed. Consequently, we conjecture that these
25 edge-preserving DoG kernels also stabilize the optimization process.

## B  Pytorch-style Pesudocode of Aliasing Reduction Module

---
**Algorithm 1** Pytorch-style Pesudocode of ARM for ViT/DeiT
---

```
# Inside the transformer block:
# B: batch size, N: token sizes , C: channel number.
# H' and W': the original spatial sizes of flattened attention maps.
# self.ARM: the proposed Aliasing Reduction Module.
###### Fold the attention back to spatial #############
attention = self.attn(self.norm1(x)) # B, N, C
attention = attention.permute(0,2,1).view(B,C,H',W') # B, C, H', W'
###### Perform anti-aliasing #############
attention = self.ARM(attention)
x = x + self.drop_path(attention.permute(0,2,3,1).view(B, N, C))
x = x + self.drop_path(self.mlp(self.norm2(x)))
return x
```
---

27

To better illustrate our simple yet effective design, we also pesudocodes in Pytorch-style. We provide both examples for two popular vision transformer structures: ViT[5] (DeiT [6]) [1] and Swin Transformer [7] [2], in Algorithm 1 and Algorithm 2.

The proposed ARM is versatile with most vision transformer families, by directly anti-aliasing the self-attention representations in the transformer blocks. As discussed in Section 3.2, the ARM operator can be chosen flexibly among a traditional low-pass filter, e.g. Gaussian filter, a learnable convolutional filter, or a pre-defined filter bank. Any mentioned choice could consistently bring some improvements to the switchable baselines.

---

**Algorithm 2** Pytorch-style Pesudocode of ARM for Swin Transformer

---

```
# Inside the Swin Transformer Block:
# B: batch size, nW: number of windows , C: channel number, H, W: the original size.
# window_size, window_reverse: the size of each window, and the function to reverse windows back.
# self.ARM: the proposed Aliasing Reduction Module.
###### Compute window attention and merge the windows#############
attn_windows = self.attn(x_windows, mask=self.attn_mask) # nW*B, window_size*window_size, C
attn_windows = attn_windows.view(-1, self.window_size, self.window_size, C)
###### Perform anti-aliasing #############
shifted_x = self.ARM(window_reverse(attn_windows, self.window_size, H, W)) # B H W C
###### Reverse cyclic shift(Omitted in pesudocode) #############
x = roll(shifted_x)
x = shortcut + self.drop_path(x)
x = x + self.drop_path(self.mlp(self.norm2(x)))
return x
```

---

## C  Downstream Task Evaluations

To better demonstrate the effectiveness of the proposed method, we further conduct evaluations on downstream tasks including object detection and semantic segmentation. We choose a strong architecture Swin Transformer [7] as our baseline.

**Object Detection.** We perform object detection experiments on COCO 2017 [8] dataset, which contains 118K images for training, 5K images for validation, and 20K images for test-dev. We consider two widely-adopted object detection frameworks including Mask R-CNN [9] and Cascade Mask R-CNN [10] in mmdetection [11]. Following [7], we keep the consistent settings including multi-scale training, AdamW optimizer (with an initial learning rate of 0.0001, weight decay of 0.05, and batch size of 16). We adopt both 1x (12 epochs) and 3x (36 epochs) schedule and similar hyperparameter settings from the open-source implementation[3].

| Method | Backbone | Pre-trained | LR Schedule | Box mAP | Mask mAP |
|---|---|---|---|---|---|
| Mask R-CNN [9] | Swin-T | ImageNet-1k | 1x | 43.7 | 39.8 |
| | **Swin-T w ARM** | ImageNet-1k | 1x | **44.8** | **40.5** |
| | Swin-T | ImageNet-1k | 3x | 46.0 | 41.6 |
| | **Swin-T w ARM** | ImageNet-1k | 3x | **46.7** | **42.1** |
| Cascade Mask R-CNN [10] | Swin-T | ImageNet-1k | 1x | 48.1 | 41.7 |
| | **Swin-T w ARM** | ImageNet-1k | 1x | **48.9** | **42.3** |

Table 1: Results on COCO object detection and instance segmentation. The baseline architecture follows Swin-T. The models integrated with our proposed ARM are shown in bold font.

From Table 1, the proposed ARM enhances both baselines consistently in 1x and 3x schedule without bells and whistles. The results verify the effectiveness of ARM on downstream tasks.

**Semantic Segmentation.** We also evaluate our method on semantic segmentation, utilizing the widely-used ADE20K [12] dataset. ADE20K covers 150 semantic classes, with 20K images for training, 2K images for testing, and 3K for testing. Following [7], UperNet [13] structure in mmsegmentation [14] is used. For training, the AdamW optimizer with an initial learning rate of $6 \times 10^{-5}$ and a weight decay of 0.01 is employed. The models are trained for 160K iterations on 8 Tesla V100 GPUs. We also adopt the consistent data augmentations in mmsegmentation

---

[1] https://github.com/facebookresearch/deit
[2] https://github.com/microsoft/Swin-Transformer
[3] https://github.com/SwinTransformer/Swin-Transformer-Object-Detection

implementation. During inference, a multi-scale testing strategy exploits [0.5, 0.75, 1.0, 1.25, 1.5, 1.75]× resolutions are exploited. We report mIoU on the validation set in Table 2.

| Method | Backbone | Crop Size | LR Schedule | mIoU |
|--------|----------|-----------|-------------|------|
| UperNet | DeiT-S | 512 | 160K | 44.0 |
| UperNet | Swin-T | 512 | 160K | 45.8 |
| | **Swin-T w ARM** | 512 | 160K | **46.9** |

Table 2: Results of semantic segmentation on ADE20K dataset. The models integrated with our proposed ARM are shown in bold font.

**Generalization towards Common Corruptions.** We provide detailed error rates towards different types of common corruptions on ImageNet-C. As mentioned above, transformers have demonstrated dominance against corruptions compared to CNNs. Likes anti-aliasing the CNNs in [1], anti-aliasing in vision transformers also upgrades the feature robustness. Moreover, we can find that while Swin-T has a overall lower error rate compared to anti-aliased ResNet-50, it acts poorly when dealing with certain corruptions such as JPEG-compression and pixelate. When integrated with our aliasing reduction module, the model gains a clear boost in robustness, particularly towards those corruptions that are not handled well by the original transformer architecture.

| | Error Rate towards common corruptions on ImageNet-C | | | | | | | | | | | | | | | norm |
|---|---|---|---|---|---|---|---|---|---|---|---|---|---|---|---|---|
| | Noise | | | Blur | | | | Weather | | | | Digital | | | | |
| | **Gauss** | **Shot** | **Inpulse** | **Defocus** | **Glass** | **Motion** | **Zoom** | **Snow** | **Frost** | **Fog** | **Bright** | **Contrast** | **Elastic** | **Pixel** | **Jpeg** | **mCE** |
| AA R50 | 63.86 | 66.07 | 69.15 | 58.36 | 71.70 | 60.74 | 61.58 | 66.78 | 60.29 | 54.40 | 31.48 | 58.09 | 55.26 | 53.89 | 43.62 | 73.73 |
| Swin-T | 52.46 | 54.42 | 54.12 | 68.31 | 83.68 | 65.52 | 72.85 | 56.91 | 52.84 | 49.02 | 47.79 | 45.50 | 75.96 | 67.03 | 64.11 | 60.7 |
| w ARM | 51.17 | 54.03 | 53.58 | 66.25 | 83.06 | 65.07 | 70.44 | 57.22 | 57.12 | 46.79 | 45.15 | 44.99 | 77.12 | 63.88 | 61.52 | 59.8 |

Table 3: Generalization towards corruptions. The error rates (lower is better) on ImageNet-C. In the first row we provide ResNet-50 with anti-aliasing in [1]. The next two rows respectively show the Swin-T's performance, and the Swin-T with our ARM module.

## D License of Dataset

**Datasets.** We use four datasets including ImageNet, ImageNet-C, MS COCO, and ADE 20K.

ImageNet [4]: BSD 3-Clause License.

ImageNet-C [5]: Apache-2.0 License

MS COCO [6]: Creative Commons Attribution 4.0 License

ADE20K [7]: Creative Commons BSD-3 License