# OpenReview forum: "Blending Anti-Aliasing into Vision Transformer"
_NeurIPS.cc/2021/Conference — NeurIPS 2021 Poster_

### Official Review · Reviewer_HeAy · 2021-07-08

**Rating:** 8
**Confidence:** 5

**Summary:**

This paper analyzes the uncharted problem of aliasing in vision transformer and explore to incorporate antialiasing properties. To handle that, they propose a simple yet effect module named Aliasing-Reduction-Module(ARM) to smooth the self-attention output. The ARM contains a set of filters to augment the attention layer output. Extensive experiment results show such approach achieves a clear gain but with a mirror GFlop increase along with more efficiency and robustness.

**Limitations And Societal Impact:**

No major weakness.
Several points which are needed:

1.  During implementation, can the ARM module directly put after the softmax computation? Since the attention maps have lots of high frequency noise, i.e. aliasing effects. It seems more straightforward to anti-alias the attention scores compared to the post-projection features.

2. The anti-aliasing operation introduces new hyperparameters before training, such as the parameters of the selected Gaussian kernels. It’s worth noting about how to choose the proper kernels for different architectures and the training stability.
Here are several detailed concerns:

Fruther suggestions :

1,  What about put both Pre-defined Filter Bank and learnable convolution(Depth-Wised)  into ARM ? (I wonder whether it have more performance boost).

2,  is input of ARM attention maps or value output of self-attention? Fig-3 looks a little confusing. I believe this is value output of self attention.

3,  Better put the figure on the top of pages.

4, Better to show out experiment results on more exsiting transformer frameworks.


**Main Review:**


1. Overall, the motivation is very good and clear. Rather to design another transformer architecture, this paper tries to solve the aliasing problems brought by the tokenization process of VIT. This is an interesting problem. Also, it is complementary to many transformer design studies.

2. The writing is good and easy to follow.

3. The design of ARM is well motivated and the design of filtering process is well explored in both method and extensive experiments. Unlike tokens in natural language that stand for separated words, the tokens in vision transformers commonly represent manually-defined windows. As image contents may vary, the pairwise relationships between these tokens may be ill-posed and contain considerably amount of high frequency noise. It’s intuitive and novel to “smooth”the “aliased interactions” between tokens. To this end, this work proposes a simple yet effective design. The experiment results across different transformer architectures also validate the effectiveness of the plug-in design. Meanwhile, the work chooses to directly smooth the attention maps, which is compatible with other advances such as incorporating convolution into the patchify stem.

4. The findings in Section 4.1 and 4.2 are useful and crucial, where the choices of filter types and placements are explored. It’s interesting to observe that anti-aliasing techniques conducted on early transformer blocks are useful while too much smoothing on deeper layers might be harmful. The authors conduct extensive comparisons and discussion to explain the reason. These conclusions also provide some insights and directions to boost the development of vision transformers.

5. The proposed methods can be easily deployed into different transformer frameworks and leads to performance gains and more robust. These are well studied in experiment part.

Rather than tricky improvements on Vision Transformer ( I am tired with several attention layer extensions with SOTA performance),  this paper points out an important and potential problem of current VIT design named  Aliasing  Effect brought by tokenization process. It also give a simple yet effective solution.  Based on all the points, I give my init rate as accept.







**Time Spent Reviewing:**

6.5

---

> ### Author Response · Authors · 2021-08-10
> **Response to Reviewer HeAy**
>
> Thank you for your detailed and constructive comments. You questions and concerns are addressed below:
>
> **Q1: Can the ARM module be directly put after the softmax computation?**
>
> **A1:** Yes, it’s straightforward to put the ARM module directly after softmax. Our choice here to anti-alias the post-projections features provides a more efficient and pluggable implementation. The reason is that in general transformer codebases such as Algorithm 1 and 2 in the Appendix, the pipeline firstly processes the images to be 3-dimensional($B \times C \times N$), followed by *self.attn* function that contains both softmax and projection. Since our smoothing module is spatially-operated on the whole attention maps($B \times C \times H \times W$).
>
> If we directly ensemble the ARM module after softmax, we need two more resize operations between 3-D and 4-D features for transformers such as Swin. We found that these two variants have similar performance and share the same FLOPs, while the post-projection one has a better efficiency(Throughput(img/s): 721.9 VS 705.3). As the projections can be regarded as matrix dot products, these two operations are mathematically equal. Therefore, we choose a more convenient and efficient implementation here.
>
> **Q2: Hyper-parameters on the proper kernels for different architectures.**
>
> **A2:** The hyper-parameters about the filter bank majorly include the number and type of pre-defined filters. For the number, we find that a relative medium size of 8 gives a good performance. When the number is smaller, more filters improve the richness of the bank as well as the performance. However, a larger size > 8 doesn’t bring further improvement. For the type of pre-defined filters, we find that the results are robust towards different kernel parameters thus we choose to randomly sample the kernels in each run, as detailedly discussed in the Appendix A. We also find that the observations hold for both ViT/DeiT, T2T-ViT, and Swin transformer.
>
> **Q3: Put both the pre-defined filter bank and learnable depthwise convolution into ARM.**
>
> **A3:** In that case, the followed depthwise convolution could be regarded as a refinement on the filtered attention maps. We’ve tried to put both two modules together and found the results are similar, leading to a 0.1-0.2% difference. Another interesting observation is that if we put more convolutions and parameters to the coefficient estimator, the performance could be further improved.
>
> **Q4: Is input of ARM attention maps or value output of self-attention?**
>
> **A4:** Your understanding is correct. As mentioned in the first question, we choose a more efficient and convenient implementation of putting the ARM to the value output of self-attention. Nevertheless, these two implementations are similar and can produce identical performances.

---

### Official Review · Reviewer_EQZA · 2021-07-15

**Rating:** 7
**Confidence:** 5

**Summary:**

This paper proposes a plug-and-play anti-aliasing reduction method to solve the aliasing problem in the vision transformer. It analyzes the effectiveness and generalization ability in multiple tasks and various vision transformer methods. Experimental results demonstrate the effectiveness of the proposed method.

**Limitations And Societal Impact:**

The limitations of the paper is not explained clearly. More experimental results should provide to analyze the limitations of the proposed methods.

**Main Review:**

This paper proposes a simple plug-and-play anti-aliasing reduction method. It analyzes the effect of the different filters (e.g., Gaussian filters, learnable convolution filter, pre-defined filter bank) on the anti-aliasing reduction and proposes an anti-aliasing reduction based on the pre-defined filter bank. The corresponding ablation studies in Section 4.1 demonstrate the effect of the anti-aliasing filters.

There are some unclear points in the paper.

The aliasing in the transformer may be caused by low-resolution of the image patches. The analysis in Section 3.1 is not clear and does not support the motivation of the paper.

It is not clear why the pre-defined filter bank performs better than the learnable convolution filter. In addition, the pre-defined filter bank needs to determine the number of the pre-defined filter dictionaries. Does the number of the pre-defined filter dictionaries affect the final results?

The contribution of the paper is marginal. For the aliasing, it is well-known that using Gaussian blur or other blur kernels can reduce such xx to some extent. As shown in Table 1. Using Gaussian filter is able to improve the performance. This is like an implementation trick. In addition, the paper does not provide insightful analysis on the proposed method.


---
The authors solve my major concerns.  It would be better to include the corresponding materials of the rebuttal in the revised paper properly.


**Time Spent Reviewing:**

8

---

> ### Author Response · Authors · 2021-08-10
> **Response to Reviewer EQZA**
>
> Thanks a lot for your precious time and important reviews. Our response to your concerns are provided below, where we also explain and recap our work.
>
> **Q1: The reason for aliasing in transformers, and the paper’s motivation.**
>
> **A1.1:** You’ve raised a very important point here: the transformer’s capability might be constrained by the scale of patches. From the signal processing perspective, a direct solution to prevent aliasing is increasing the "sampling rate". However, there always exists a tradeoff between the token sizes/scales and the computation efficiency. For the well-established Vision Transformer(ViT) family that has quadratic complexity on pairwise computations, it’s computationally infeasible to ease the aliasing issue from providing higher-resolution tokens.
>
> With the booming development of vision transformers, the architecture can now be divided into a patchify stem and the following block-wise transformer encoders. We’ve mentioned in Section 3.1 that the aliasing issue comes from the aggressive downsampling in the patchify process, as well as the attention operations. For any vision transformer architectures, bringing in larger resolution inputs and adequate tokens could help ease the aliasing phenomenon. Nevertheless, the token-wise self-attention regime in recent vision transformers makes the aliasing issue inevitable. Therefore, we choose to produce a fix inside the transformer encoder. We want to emphasize that our design is compatible with other improvements that are made in the input and the patchify stem.
>
> **A1.2:** To further explain the aforementioned aspect, we perform several additional experiments. To investigate the effect on larger inputs, we train another SwinT with $384 \times 384$ input on ImageNet-1k, as well as our ARM variant. We can clearly find that our module brings improvement when the inputs are larger.
>
> | Model           | Input size |Top-1 Acc |
> | --------------- | ------------------ |------------------ |
> | Swin-T    | $224 \times 224$ |      81.2       |
> | **Swin-T w Ours**    | $224 \times 224$ |  **82.0**           |
> | Swin-T    | $384 \times 384$ |     82.1        |
> | **Swin-T w Ours**   | $384 \times384$ |   **82.8**          |
>
>
> We also evaluate the transformers that use different sizes of tokens. You can see that by splitting the inputs to smaller tokens($16 \times 16$  -> $8 \times 8$ ), the transformer obtains more tokens, better performance, as well as much more complexity cost(inference time). When employed with our module, both models receive a clear improvement with only a negligible computation overhead. Moreover, our module helps the DeiT-S/16 obtain the similar performance as DeiT-S/8, of which is four times heavier. Compared to naively increasing the resolutions, these results indicate that our approach is a more efficient way of mitigating the aliasing effects.
>
> | Model           | Token Size |  Token number  | Speed(image/s) |Top-1 Acc |
> | --------------- | ------------------ |------------------ |------------------ |------------------ |
> | DeiT-S/16    | $16 \times 16$ |      196       |	425.6    |    79.8   |
> | DeiT-S/8    | $8 \times 8$ |     784        |   102.1      |   80.9      |
> | **DeiT-S/16 w Ours**    | $16 \times 16$ |   196       |	  405.8     | **80.7**         |
> | **DeiT-S/8 w Ours**   | $8 \times8$ |   784       |     98.5     |  **81.6**      |
>
> **Q2: Does the number of pre-defined filters affect the final results?**
>
> **A2:** We’ve discussed how the varying sizes influence the results in the paper, Table 5 and L291-L296. We’ve experimented with different sizes $n = 2, 8, 16, 24$ and found a trend of improvements from larger banks when the number is small. A larger dictionary with more than 8 filters doesn’t bring further improvement. The details are provided in the paper.
>
> **Q3: The reason why the filter bank strategy performs better than convolution filter.**
>
> **A3:** Thanks for bringing up this concern. Compared to the normal feature maps, self-attention maps in transformers vary across different images and heads, containing much more high frequency parts. We evaluate the variance of input features versus their attention maps. variance: $0.14$ vs $0.39$. You can see that attention representations have a much higher variance. Meanwhile, different attention heads may contain totally different regional information. It’s extremely difficult to learn the weights spatial convolution and hard to optimize. In contrast, the low-rank constraints on the generated filters provide smoothed candidates of these high-frequency attention maps. Then it’s much easier for the spatially-variant coefficient estimator to learn the combination weights. From our perspective, the aliasing reduction on attention maps requires more redundant smoothness choices, while it’s easy to only obtain non-sense features and poor convergences using only convolution. That’s the reason why the filter bank performs better.
>
>
> **Q4: Contribution of this paper and detailed analysis.**
>
> **A4.1:** Firstly, we would like to emphasize that the performance in Table 1 exploits our conclusion about the placements in Sec. 4.2. In Table 2 and Table 3, you can see that the position of filtering is crucial. In fact, directly ensembling Gaussian or other filters is harmful and would bring 2% to 3% decrease in accuracy. We conducted extensive experiments and found that this finding holds for all three types of ARM design, which is non-trivial since there exists no previous discussion about the difference about placements. Besides, while the ``blurpool’’ proposed by Zhang et al[1] consistently improves CNNs’ performance, it doesn’t work for vision transformers as we’ve discussed in Sec. 4.2. Instead of being an implementation trick, our finding is beneficial for further research on optimizing the vision transformers.
>
> **A4.2:** Secondly, your concern about filtering being well-studied is correct. From our perspective, while the problem of aliasing and blurring kernels have been widely explored in low-level and signal processing, this topic has never been discussed in vision transformers. Note that due to the vast development of ViT, the transformer pipeline becomes much more complex. Numerous efforts are being made to patch projection[3] and the encoder architectures, while the token-wise attention operation has been a long-standing overlooked term. Without modifying the general pipeline, our work makes attempts to process the attention maps instead and is compatible with other well-established designs. Also from Q1, you can see that our module is an efficient choice with different scales. We believe that more classic knowledge is still valuable and unnoticed for providing much help.
>
> **A4.3:** Thirdly, the problem of aliasing is a traditional problem, but has begun to catch attention in modern deep learning structures recently.  Anti-aliased models could exhibit better stability and translation equivariance. We’ve also discussed the generalization towards corrupted inputs in Appendix.C. Note that due to space limits, we put some detailed analysis about the ``serial’’ characteristics of vision transformers to the Appendix.D. This section explains our placement strategy. We will also include more detailed discussion, as well as recent progress[2,4] on transformers that also support our findings .
>
> We really appreciate your valuable comments. Hope the response we provided could solve some of your concerns. We will also take your suggestions to polish our draft.
>
>
> Reference:
>
>
> [1] Richard Zhang. "Making convolutional networks shift-invariant again." ICML 2019.
>
> [2] Bhojanapalli, Srinadh, et al "Understanding robustness of transformers for image classification." arXiv preprint 2021.
>
> [3] Tete Xiao, Mannat Singh, Eric Mintun, Trevor Darrell, Piotr Dollár, Ross Girshick.  "Early Convolutions Help Transformers." arXiv preprint 2021.
>
> [4] Yihe Dong, et al "Attention is Not All You Need: Pure Attention Loses Rank Doubly Exponentially with Depth." ICML 2021.

---

> ### Author Response · Authors · 2021-08-31
> **Request for potential feedback on the rebuttal**
>
> Dear Reviewer EQZA,
>
> We appreciate your time for reviewing, and we really want to have a further discussion with you to see if our response solves the concerns. Your initial concerns are about our contribution, especially the aliasing problem in vision transformers. We have demonstrated the investigations on larger scaled inputs and higher sampling rates(token sizes), and we believe the consistent effects could better explain the necessity of re-analyzing the aliasing problems in vision transformers, as well as the effectiveness of ARM.
>
> We have also addressed other thoughtful questions raised by the reviewers, including novelty and performance. For instance, reviewer fmAF mentioned about the quantifiable metrics about anti-aliasing, which provides more reliable measurement of ARM’s properties. We also provide visualization comparisons by ensembling our strategy to SOTA semantic segmentation models in response to Reviewer yhPZ, from which you can clearly see the improvements on the small objects and contours(important hardcases/bottlenecks of semantic segmentations).
>
> We really hope that our work’s impact is better conveyed and highlighted with our feedback. It would be great if the reviewer can kindly check our responses and provide feedback with further questions/concerns (if any). We would be more than happy to address them. Thank you!
>
> Best wishes,
>
> Authors

---

> > ### Comment · Reviewer_EQZA · 2021-09-01
> > **Reply**
> >
> > Dear Authors,
> >
> > Thanks for clarifying the motivation and the contributions of the paper. My major concerns have been solved. It would be better to include the corresponding materials of the rebuttal in the revised paper properly.

---

> > > ### Author Response · Authors · 2021-09-02
> > > **Reply**
> > >
> > > Dear reviewer EQZA,
> > >
> > >
> > > Thank you for finding our rebuttal helpful! We are glad to know that we have resolved your major concerns, and we will include the corresponding materials of the rebuttal in the final revision. We were wondering if you would like to update your ratings, given the discussion phase is nearing its end. Thank you very much for your precious feedback, we really appreciate it.
> > >
> > >
> > > Best Regard,
> > >
> > > Authors

---

### Official Review · Reviewer_fmAf · 2021-07-16

**Rating:** 6
**Confidence:** 3

**Summary:**

The paper discussed the aliasing effects for vision transformers and proposed a simple yet effective fix to it — applying anti-aliasing filters on the attention maps. The authors demonstrate such an improvement brings 0.5% - 1% performance gain on ImageNet classification tasks over two different vision transformer architectures.


**Limitations And Societal Impact:**

Yes

**Main Review:**

Pros:
* Nice finding. I had similar concerns on ViT families to split the image into non-overlapped patches. It's good to see a minor fix could boost performance.
* Thorough experiments. All the experiments deliver useful messages and come with a good analysis of the experimental results.
* The presentation is clear. It's not hard to follow the paper.


Cons:
* There's a lack of quantifiable measures on the aliasing effects. Besides, there are also not many qualitative examples for attention maps before and after anti-aliasing. As a result, it's not 100% convincing to me whether the proposed improvement and the performance gain it brought is due to "anti-aliasing".
* I assume this effect might be more severe on dense prediction tasks, potentially resulting in block effects on the feature map. Have you considered applying the approach to those models?
* There's no discussion on whether a non-patch-based transformer would have similar issues, e.g., non-local blocks.


Table 2: How do you apply ARM on patch embeddings? This is not straightforward to me. Have you considered adding an anti-aliasing feature to the input feature map before any embedding computation?
In Table 2, which filters setting do you chose? Will the placement strategy's relative performances vary for different filter settings?


You should be able to generate an attention map at a much higher "sampling rate" during training through overlapped image blocks followed by anti-aliased downsampling. Would anti-aliased downsampling on that high-res attention map get the ideal attention map with downstream performance gain? If yes, how about learn a filter to mimic that attention instead?




Table 3: The proposed approach seems to work only for the first layer. It is not clear whether this finding is generic to all different transformer models. And the author's justification for the experiments is not very convincing. Do you think Table 3's finding (only applying early layers matters) is generic for all ViT families and why? Have you visualized the attention maps to justify your conjectures?


Do you maintain the attention maps magnitudes through the filtering/smoothing operations?


The choice of the filter bank is quite adhoc. What are the other fixed filterbank options you have tried? How about DCT, or wavelets? Compared to fully learnable filters options, is this choice of fixed filterbank matter or the fact that the filters are low-rank matter? How about do learnable low-rank 5x5 filters?




Is there any way to quantify the aliasing effects? How to convince readers the performance improvement brought by the proposed approach is due to the anti-aliasing effect? The author should consider building a stronger connection between aliasing effects and different discussed options, e.g., through spectral analysis of the attention map or measuring some form of invariance or equivariance. For instance, the Zhang et al. ICLR19 paper [9] measures the amount of shift-invariance of CNNs w/o anti-aliasing convs. No quantifiable metrics for aliasing is a primary concern for me as there could be other explanations why adding a conv filtering on the attention map could help improve tasks performance.

**Time Spent Reviewing:**

5

---

> ### Author Response · Authors · 2021-08-10
> **Response to Reviewer fmAf**
>
> We appreciate the thorough comments and sincerely thank you for comprehensively understanding from multiple aspects.
>
> Our response to the individual concerns are given below:
>
> **Q1:Quantifiable metrics for aliasing.**
>
> **A1:** Quantifiable evaluation is indeed an important metric to evaluate whether our ARM module brings ``anti-aliasing’’ property and improves the shift-invariance. In fact, we conducted quantitative evaluation of the aliasing effects using the ImageNet-C dataset in the Appendix Table.3. Since the dataset contains various common corruptions on images, the error rates on ImageNet-C demonstrate the generalization and stability towards corruptions. This metric was also used by Zhang et al as one aliasing testbed.(Note that the AA-R50 in Appendix Table.3 denotes the anti-aliasing ResNet-50 proposed in [1]). We can find that our module upgrades the robustness and stability, leaving a clear margin towards the anti-aliased CNN that shares similar computation costs and parameter amounts. We really appreciate your suggestion and agree that the quantifiable metrics are important. Thus we will reorganize the experiment structures by putting this section in the main paper. Moreover, we additionally include another metric you’ve mentioned: consistency, used in [1] [2] to measure the translation equivariance. The elevation in consistency further supports that our module upgrades the model consistency, besides the task performance.
>
> | Consistency[1]           | Baseline | Ours  |  Delta  |
> | --------------- | ------------------ | ------------ | ------------ |
> | DeiT-S     |       90.5        |	**91.7**	|   +1.2	|
> | Swin-T |      91.2     |    **92.3**	|	+1.1  |
>
> **Q2: Apply the module to dense prediction models.**
>
> **A2:** We have validated dense prediction tasks in the Appendix based on the Swin-T and UperNet baseline(Appendix Table 2). In addition, we further apply our module to SETR[8]. You can see that both Swin-UperNet we already provided and the newly-added SETR are improved.
>
>
> | Model           | mIoU(ms+flip) |
> | --------------- | ------------------ |
> | SETR-MLA     | 49.0              |
> | **SETR-MLA w Ours** | **49.5**          |
>
> **Q3: Discussion on a non-patch-based transformer.**
>
> **A3:** This is an interesting suggestion. As we are focusing on the vision transformer family in this work, we didn’t consider these non-patch-based models that use point-to-point relations. Hereby we try to apply our module to the non-local networks. We use the open-sourced implementation of GCNL[3]. Surprisingly, we observe that the performance is further improved (From 77.4 to **77.9**). An initial guess is that since pairwise relationships contain a considerable amount of high frequency, the module smooths the noisy features and eases the training process. It’s an interesting finding which we will dive deeper into.
>
> **Q4: Details about applying ARM on patch embeddings, filter settings, and placements.**
>
> **A4.1:** We conduct experiments on anti-aliasing the patch embeddings in Table 2 by smoothing/blurring the downsampled representations after patch embeddings. ($B \times C\times H \times W$ ----> $B \times C\times H/k \times W/k $ ----> ARM).  This practice follows the anti-aliasing strategy proposed by Zhang et al.[1], which is also the traditional way of anti-aliasing in signal processing. These experiments are aimed at investigating whether direct anti-aliasing on downsampled representations provides improvements. From Table 2, we eventually find that the ``blurpool’’ practices in CNNs don’t hold for vision transformers.
>
>
> **A4.2:**  *Have you considered adding an anti-aliasing feature to the input feature map before any embedding computation?*
>
> Yes, we’ve experimented with adding an anti-aliased feature to input during the patch projection process. We find that this practice improves neither the accuracy nor the consistency metrics in Q1. This phenomenon implies that fixing the patch embedding is inadequate to address the aliasing issue in transformers, motivating us to focus on the attention operation instead.
>
> **A4.3:** In Table 2, we report the results using the pre-defined filter bank setting. We’ve tried all three filter types combinations with different placements and found that the concluded placement strategies share across different filter settings.
>
> **Q5: Would Higher Sampling Rate help and how about learning to mimic the attention instead?**
>
> **A5:** Yes, a higher sampling rate through overlapping blocks or smaller token sizes can provide more fine-grained attention maps, as well as more tokens. Anti-aliasing downsampling on higher-res attention maps generally brings performance gain. However, there always exists a trade-off between the "sampling-rate" and computation costs, especially from the quadratic complexity of self attention.
> From our perspective, learning a conv/filter to mimic the higher-res attention is interesting and feasible. There are several previous works that exploit the teacher-student knowledge distillation regime, such as attention distillation[4,5]. These works tried to mimic the better or higher-level attention maps from the teacher network. However, there exists little progress on "self-mimicking" the ideal/higher attention maps to lower levels with no teacher. We believe that’s an interesting topic to study.
>
> **Q6: Is Table 3's finding (only applying early layers matters) generic for all ViT families and why?**
>
> **A6:** Yes, we think Table 3's finding (only applying early layers matters) is generic for various vision transformers. We’ve experimented with DeiT, Swin, and T2T-ViT architectures, which all exhibit similar observations. By applying the ARM module to early layers, e.g. 1/4 , these models receive consistent improvements. Meanwhile, some recent progress also supports this finding[6,7]. Besides the explanation we made in Appendix.D, we will add more discussion about the reason.
>
> **Q7: Do you maintain the attention maps magnitudes?**
>
> **A7:** Yes, the magnitudes are maintained through smoothing operations.
>
> **Q8: The choice about the filter bank.**
>
> **A8:** As mentioned in Appendix. A(L22-25), we’ve found that the results are quite stable towards different random seeds. This finding conforms to the observations in computational photography. We agree that DCT and wavelet-based methods might also work. It’s interesting to study directly from the signals’ perspective. As the redundant filter bank provides enough representation ability, the low-rank filters make the optimization process much easier. Both the fixed filter bank and low-rank constraints matter. In our experiments, $5 \times 5$ and  $7 \times 7$ filters produce similar performance with  $3 \times 3$.
>
>
> Reference:
>
>
> [1] Richard Zhang. "Making convolutional networks shift-invariant again." ICML 2019.
>
> [2] Xueyan Zou, Fanyi Xiao, Zhiding Yu, and Yong Jae Lee. "Delving Deeper into Anti-Aliasing in ConvNets." BMVC 2020.
>
> [3] Kaiyu Yue, et al "Compact Generalized Non-local Network." NIPS 2018.
>
> [4] Sergey Zagoruyko, Nikos Komodakis.  "Paying More Attention to Attention: Improving the Performance of Convolutional Neural Networks via Attention Transfer." ICLR 2017.
>
> [5] Yige Li, et al "Neural Attention Distillation: Erasing Backdoor Triggers from Deep Neural Networks." ICLR 2021.
>
> [6] Bhojanapalli, Srinadh, et al "Understanding robustness of transformers for image classification." arXiv preprint 2021.
>
> [7] Yihe Dong, et al "Attention is Not All You Need: Pure Attention Loses Rank Doubly Exponentially with Depth." ICML 2021.
>
> [8] Sixiao Zheng, et al "Rethinking Semantic Segmentation from a Sequence-to-Sequence Perspective with Transformers." CVPR 2021.

---

> ### Author Response · Authors · 2021-08-31
> **Request for potential feedback on the rebuttal**
>
> Dear Reviewer fmAf,
>
> We appreciate your time for reviewing and giving valuable comments. And we would like to have a further discussion with you to see whether the response solves your concerns. You’ve raised many pivotal points, such as the quantifiable metrics for aliasing and implementation details. We’ve addressed these thoughtful questions raised by you. With our response, we really hope that the impact of our work can be better highlighted. It would be great if the reviewer can kindly check our responses and provide feedback with further questions/concerns(if any), which we would be more than happy to address.
>
> Thank you!
>
> Best Regard,
>
> Authors

---

### Official Review · Reviewer_yhPz · 2021-07-25

**Rating:** 6
**Confidence:** 4

**Summary:**

The paper proposes an Aliasing Reduction Module (ARM) consisting of an anti-aliasing filter and an external smoothing module. The paper further iterates that the module is lightweight and compatible with most existing transformer architectures. Furthermore, the method shows comparisons between a naive Gaussian filter, a Learnable Convolution Filter, and the proposed Pre-defined Filter Bank. Experiments are performed to show that placing the anti-aliasing module after the Self-Attention block yields the best results. Additionally, the method illustrates that placing the anti-aliasing module in earlier layers translates to the performance boost while inserting the module at deeper stages leads to reduced performance.

**Limitations And Societal Impact:**

The authors have sufficiently discussed.

**Main Review:**

- The paper is easy to understand.
- Results are shown on various Vision Transformers, i.e., CoaT, DeIT, and SwinT.
- Results are shown on multiple high-level vision tasks of Image Classification, Object Detection, and Semantic Segmentation.

I have the following questions:

- It is a very common practice to use overlapping patch embedding [a, b]. Would these architectures suffer from aliasing? Did you try employing your module in these algorithms?
- The performance of the proposed ARM is demonstrated across various transformers with 224x224 image size. It would be interesting to study the impact of ARM on bigger image sizes (e.g. 384x384) where the jagged effect might occur less frequently.
- Most recent methods such as [c, d] use smaller patch sizes at the patch embedding stage. However, no comparisons are made between these algorithms with the proposed method on alleviating aliasing in attention maps and their performance.
- For Object Detection and Semantic Segmentation, it would be interesting to see visual examples where baseline methods show poor performance and when the proposed ARM is employed, the results are improved.
- SwinT shifted window version provides the best results for Semantic Segmentation on ADE20K dataset. In this paper, why is the sliding window variant of SwinT used in the experiments?
- The method should elaborate on how the proposed module is different from the previously used module in [42].
- The method hypothesizes that the jagged artifacts occur due to the extraction of non-overlapping patches at the patch embedding stage. The visualization is shown for the same in Figure 1. However, the method does not show any visualization for algorithms such as [7] which use overlapping patches. As stated in Lines 32-34, does the occurrence of aliasing decreased?
- In Section 3.2, the method gives a description of the Gaussian filter, a Learnable Convolution Filter, and the proposed Pre-defined Filter Bank. The effects of each of them on a baseline are shown in Table 1. However, no visual comparison is shown on attention maps for the three methods to describe how well the three algorithms perform.
- In Figure 1, the visualization of attention maps are shown for the DeiT-B architecture. Would this visualization differ for different architectures like SwinT and CoaT etc?

[a] Haiping Wu, Bin Xiao, Noel Codella, Mengchen Liu, Xiyang Dai, Lu Yuan, and Lei Zhang. Cvt: Introducing convolutions to vision transformers. arXiv 2021.

[b] Ali Hassani, Steven Walton, Nikhil Shah, Abulikemu Abuduweili, Jiachen Li, and Humphrey Shi. Escaping the big data paradigm with compact transformers. arXiv 2021.

[c] Kai Han, An Xiao, Enhua Wu, Jianyuan Guo, Chunjing Xu, and Yunhe Wang. Transformer in transformer. arXiv 2021.

[d] Chun-Fu Chen, Quanfu Fan, and Rameswar Panda. Crossvit: Cross-attention multi-scale vision transformer for image classification. arXiv 2021.

* * * * *
### **Post Rebuttal Comments**

The authors have responded to my concerns with thorough experiments. The proposed method provides favorable results. Therefore, I am also leaning towards acceptance of this paper. Since the authors showed in rebuttal that the overlapping patch embedding models also suffer from aliasing, I would suggest including these findings in the paper.

**Time Spent Reviewing:**

16

---

> ### Author Response · Authors · 2021-08-10
> **Response to Reviewer yhPz**
>
> **Q1: Do overlapped patch embeddings suffer from aliasing?**
>
> **A1:** Yes, we’ve tried to employ some overlapping patch embedding models with our module and found that the problem of aliasing also existed. As originally mentioned in our paper, T2T-ViT in Table 4 utilizes the soft patch splits with overlapping strategy. Nevertheless, you can see that our method still yields improvement on it.
>
> Note that when we were working on this paper, few transformers that exploit overlapping windows were available or open-sourced (including the [1,2] you’ve mentioned). To further support the observations, we employ our method to some recently-available transformers CCT[2] and PVTv2[3]. We can see that both of them receive performance boosts. The results indicate that transformers with overlapped tokens still suffer from aliasing and our module could produce a fix.
>
> | Model        |Top-1 Acc |
> | ---------------|------------------ |
> | PVT-V2-B1	  |     78.7        |
> | **PVT-V2-B1 w Ours**   |  **79.9**          |
> | CCT-14   |     80.0        |
> | **CCT-14 w Ours**   |  **80.7**          |
>
> **Q2: The impact of ARM on bigger image sizes.**
>
> **A2:** Thanks for the interesting suggestion. To study that, we train another SwinT with $384 \times 384$ input on ImageNet-1k, as well as its ARM variant. The validation accuracy on ImageNet-1k is Swin-T: $82.1$ VS Swin-T+Ours: $82.9$. We can see that bigger input sizes still suffer from aliasing. According to the Nyquist-Shannon sampling theorem, the aliasing issue occurs when the sampling rate is lower than 1/2. From this perspective, the problem still exists for larger image sizes due to the downsampling.
>
> | Model           | Input size |Top-1 Acc |
> | --------------- | ------------------ |------------------ |
> | Swin-T    | $384 \times 384$ |     82.1        |
> | **Swin-T w Ours**   | $384 \times384$ |   **82.8**          |
>
>
> **Q3: Comparison Between methods with smaller patch sizes.**
>
> **A3:** During our experiments, we’ve found that using different patch sizes in the same transformer leads to different computation costs and performance. Since TNT and CrossViT you’ve mentioned are not officially open-sourced until now, we choose to compare DeiT with two different patch sizes. Using a smaller patch size leads to relatively better performance, while bringing in heavier computation costs. There always exists a trade-off between the patch size and efficiency. However, you can see that our module consistently improves the smaller patch version of DeiT. Moreover, our module helps the DeiT-S/16 obtain the similar performance as DeiT-S/8, of which is four times heavier. We believe it’s also an interesting direction to study the trade-off.
>
> | Model           | Token Size |  Speed(image/s) |Top-1 Acc |
> | --------------- | ------------------|------------------ |------------------ |
> | DeiT-S/16    | $16 \times 16$ |      425.6    |    79.8   |
> | **DeiT-S/16 w Ours**    | $16 \times 16$ |     405.8     | **80.7**         |
> | DeiT-S/8    | $8 \times 8$ |     102.1      |   80.9      |
> | **DeiT-S/8 w Ours**   | $8 \times8$ |      98.5     |  **81.6**      |
>
>
> **Q4: Visual examples for segmentation or detection, when the proposed ARM is employed, the results are improved.**
>
> **A4:** Thanks for this nice suggestion! We provide some semantic segmentation results with and without our ARM module in this [Anonymous Link](https://anonymous.4open.science/r/seg_visualization-E03E/README.md). We didn’t do cherry picks. You can clearly see that the outputs with our module perform better, especially on the small objects and contours. This phenomenon closely relates to the issue of aliasing.
>
>
> **Q5: The SwinT model used for Semantic Segmentation on ADE20K.**
>
> **A5:** We didn’t use the sliding window version of Swin-T. In practice, we directly used the Swin-T model provided by Swin Transformer’s official codebase(https://github.com/SwinTransformer/Swin-Transformer-Semantic-Segmentation). The model they provided uses the shifted window version. You can see from the website that the model has 45.8 mIoU. We didn’t modify the structure.
>
> **Q6: Elaborate on how the proposed module is different from BLADE[4].**
>
> **A6:** Our model is distinctive from the design in three perspectives.
>
> Firstly, our ARM module is designed to overcome the aliasing problem caused by the aggressive downsampling and block wise operations in vision transformers, while BLADE aims at general enhancement problems in computational photography.
>
> Secondly, unlike the selected edge-aware filters in [4] that are designed to produce sharp output, our module conversely targets at providing more smoothness to the high frequent attention maps.
>
> Thirdly, compared to the filters operated on the image-level in [4], our ARM module is depth-wise and works on the attention maps. Hence, filter bank choice in ours doesn’t require the finely filter selection in [4]. In contrast, it consists of much more efficient and smaller kernels and is feasible with combinations of low pass filters.
>
> **Q7: Does the occurrence of aliasing decrease for overlapping patches such as T2T-ViT? Would the visualization differ for Swin and CoaT.**
>
> **A7.1:** I put these two questions together since they share the same issue. The reason we show the visualization of DeiT is that it exploits global, standard self-attention. These properties make the attention maps of DeiT look global and visually-satisfying, reflecting more details from the input images. However, we found it difficult to visualize the attentions of some ViT variants like T2T-ViT(uses overlapping windows), swin(uses shifted, local attention), and CoaT(uses conv-attention). Their multi-head attention maps are abstract and noisy, making it hard to spot the difference in values. Thus we choose to visualize using the global architecture of DeiT.
>
> **A7.2:** In addition, we propose a quantifiable metric to test whether the aliasing issue in these "abstract" models decreases when integrated with our ARM module. By estimating the variance of output attention representations with/without our module, we can infer whether the aliasing problem is mitigated. From the table, we can see that all four models have a lower variance on their attention representations with our module. These results indicate that besides DeiT, other three "abstract"  models are also smoothed, reducing the aliasing effects.
>
> | Variance        | Baseline |	with Ours	|
> | ---------------|------------------ |------------------ |
> | DeiT  |     0.39       |    0.31        |
> | T2T-ViT  |  0.73          |    0.61       |
> | CoaT  |     0.61       |    0.52        |
> |  Swin-T  | 0.69          |   0.59        |
>
> **Q8: Visual Comparisons on the three different filters.**
>
> **A8:** Thanks for the suggestions! We will add more visual comparisons and discussion on the different effects of the three filters.
>
>
> [1] Haiping Wu, et al "Cvt: Introducing convolutions to vision transformers." arXiv preprint 2021.
>
> [2] Ali Hassani, et al "Escaping the big data paradigm with compact transformers." arXiv preprint 2021.
>
> [3] Wenhai Wang, et al "PVTv2: Improved Baselines with Pyramid Vision Transformer." arXiv preprint 2021.
>
> [4] Pascal Getreuer, et al "BLADE: Filter Learning for General Purpose Computational Photography." ICCP 2018.

---

> ### Author Response · Authors · 2021-08-31
> **Request for potential feedback on the rebuttal**
>
> Dear Reviewer yhPZ,
>
> We appreciate your time for reviewing and providing valuable comments. You’ve raised lots of thoughtful questions including the overlapping regime of patch embedding, larger image sizes or smaller patch sizes. And we really want to have a further discussion with you to see if our response solves your concerns. We’ve addressed all the thoughtful questions raised by the reviewer. We have also provided the visualizations on semantic segmentation suggested by you, which we found interesting to explain the nice property of ARM.
>
>
> We hope that our work’s contributions are better explained and highlighted. It would be great if the reviewer can kindly check our responses. If there exists any feedback with questions/concerns, we would be more than happy to address them. Thank you!
>
> Best Regard,
>
> Authors

---

> > ### Comment · Reviewer_yhPz · 2021-09-02
> > **Feedback on Rebuttal**
> >
> > Dear Authors,
> >
> > Thank you for addressing my concerns in the rebuttal. I have updated my rating and added some comments.

---

### Decision · Program_Chairs · 2021-09-27

**Decision:**

Accept (Poster)

**Comment:**

This paper initially receives mixed reviews. The authors successfully convince the reviewers with the rebuttal. The reviewers discuss extensively the strengths/weakness of this work and in the end all are in favor of accepting this paper. The area chairs agree with the reviewers' recommendation to accept this paper.